# Age-Related Response to Remimazolam among Older Patients Undergoing Orthopedic Surgery: A Single-Center Prospective Observational Study

**DOI:** 10.3390/medicina60101616

**Published:** 2024-10-02

**Authors:** Min Suk Chae, Nuri Lee, Hyun Jung Koh

**Affiliations:** Department of Anesthesiology and Pain Medicine, Seoul St. Mary’s Hospital, College of Medicine, The Catholic University of Korea, Seoul 06591, Republic of Korea; shscms@catholic.ac.kr (M.S.C.); snhkry55@naver.com (N.L.)

**Keywords:** flumazenil, general anesthesia, geriatric patients, remimazolam, general anesthesia

## Abstract

*Background and Objectives*: Remimazolam, an ultra-short-acting benzodiazepine, is increasingly used in procedural sedation and general anesthesia. It is characterized by rapid onset of action, inactive metabolites, no delay in recovery, and few adverse events. Its hemodynamic and respiratory stability are comparable to other anesthetics, and it is safe in high-risk and geriatric patients. *Materials and Methods*: This prospective, observational study enrolled 110 geriatric patients (aged 65 to 85 years) scheduled for primary total knee arthroplasty (TKA). The patients were divided into the old (65 to <75 years; *n* = 52) and the elderly (75 to 85 years; *n* = 47) geriatric groups. All surgical and anesthetic methods were applied in the same manner, and TKA was performed by one surgeon. Remimazolam was infused at 6 mg/kg/h for 3 min and then at 1 mg/kg/h until the end of surgery The primary study endpoint was the requirement for flumazenil; secondary endpoints were the times to reach a bispectral index (BIS) < 60 and >80, as well as the rate of apnea occurrence. *Results:* Flumazenil administration was similar in both groups. There were no differences in the time to reach BIS < 60 or the rate of apnea occurrence. Recovery characteristics, including the time to reach BIS > 80 and the achievement of full consciousness, were also comparable between the groups. *Conclusions:* Remimazolam is well-tolerated in geriatric patients undergoing orthopedic surgery, with minimal age-related differences in response. These results suggest that remimazolam is an appropriate anesthetic for geriatric patients, even with similar dosing strategies. It provides effective anesthetic depth with no significant increases in adverse outcomes during orthopedic surgery.

## 1. Introduction

Remimazolam is a new ultra-short-acting benzodiazepine with a pharmacology similar to other benzodiazepines. It combines the pharmacokinetics of remifentanil with the pharmacodynamics of midazolam, resulting in rapid conversion to inactive metabolites immediately after administration. The American Society of Anesthesiologists (ASA) Monitor introduced remimazolam as a noteworthy enhancement in anesthesia that practitioners should familiarize themselves with before use [1]. Like midazolam, remimazolam does not cause injection-site discomfort and can be reversed by flumazenil. Similar to remifentanil, remimazolam undergoes rapid ester hydrolysis into an inactive metabolite, avoiding accumulation during infusions and minimizing delayed recovery [2]. Despite these advantages, remimazolam should be used cautiously in combination with opioids due to the potential for profound respiratory depression.

Studies have examined the induction and maintenance of general anesthesia and procedural sedation with remimazolam, which offers a fast onset, short predictable duration of action, quick recovery time, rare metabolite accumulation, and fewer serious side effects than other benzodiazepines or currently used anesthetics. Most studies have compared remimazolam with agents such as propofol, focusing on the hemodynamic response and emergence agitation, with the findings indicating superiority or non-inferiority [3,4,5,6,7,8]. Notably, recent research has shown that remimazolam-induced anesthesia may lead to the development of tolerance [9], potentially limiting its use. Remimazolam is well-tolerated and effective for procedural sedation. It is non-inferior to propofol for general anesthesia in terms of safety and efficacy, although it induces loss of consciousness more slowly than propofol. Typically, it is recommended that drugs be administered to elderly patients at reduced doses due to the physiological changes associated with aging. Although some reports suggest that remimazolam at lower doses is effective for maintaining anesthesia and facilitating recovery in elderly patients [10], few reviews have assessed remimazolam for procedural sedation in elderly patients. They generally have compared its effectiveness with anesthetic agents such as propofol [11,12,13]. Considering the rapid increase in the elderly population, which is projected to exceed 20% of the total population by 2024, there is a growing need to re-evaluate age-related criteria for managing elderly patients, particularly in the context of an aging society. It has become crucial to subdivide elderly populations into more specific age groups to better understand the clinical implications of aging. For instance, 75 years of age has been identified as a key milestone in elderly classification [14].

Current research on remimazolam is expanding across patient groups and anesthetic settings. However, few studies have directly compared age-related differences in the response to remimazolam under general anesthesia in elderly patients. Therefore, this study investigated the potential differences in drug responses between younger (<75 years) and older (≥75 years) elderly groups under general anesthesia, using an established age classification [14], to identify age-related factors that may influence these responses.

## 2. Methods

### 2.1. Ethical Considerations

This prospective, observational study was approved by the Ethics Committee of Seoul St. Mary’s Hospital (KC22MISI0412) on 26 August 2022, adhering to the principles of the Declaration of Helsinki. This study was registered with the Clinical Research Information Service, Republic of Korea (http://cris.nih.go.kr, KCT0009464) on 24 May 2024. Informed consent was obtained from all participants before enrollment. Participants were recruited between 2 May 2023 and 30 May 2024.

### 2.2. Study Population

This study included adults aged 65 to 85 years who were scheduled for primary total knee arthroplasty (TKA). The patients were divided into two groups, namely the old group, consisting of patients 65 to under 75 years, and the elderly group, consisting of patients 75 to 85 years. For inclusion, patients were required to have an ASA physical status of I or II, ensuring that only patients with mild-to-moderate systemic disease were included. Patients with a higher ASA physical status (III or more), indicating more severe systemic disease, were excluded to minimize confounding factors related to comorbidities. Other exclusion criteria included no previous history of TKA in the affected knee, revision surgery, or simultaneous bilateral TKA. Additional exclusion criteria were a history of allergy to benzodiazepines, uncontrolled cardiovascular conditions, refusal to participate, emergency surgery, liver or psychiatric disorders, concurrent use of benzodiazepines or tricyclic antidepressants, cognitive dysfunction, and the use of medications affecting sleep. The study was terminated early in cases where side effects were predicted or occurred during anesthesia induction or maintenance. Furthermore, patients who changed their minds after providing informed consent and signing the study agreement were allowed to withdraw from the study at any time.

Based on the inclusion criteria, 52 participants were included in the ‘old group’ (65 to <75 years), whereas 47 participants were included in the ‘elderly’ group (75 to 85 years) (Figure 1).

### 2.3. Study Protocol

The same surgeon performed TKA using the same technique for all patients; medical staff assisting with the surgery followed the same protocols for all processes.

The patients underwent surgery according to their regular schedule, and all received the same anesthetic method. To confirm the patient’s condition according to remimazolam administration, time was measured during the induction and extubation period. During this process, the time at which the bispectral index (BIS) exceeded 80 was checked, and a train of four (TOF) was performed to confirm the patient’s full recovery. Extubation was performed when the patient recovered self-respiration with a TOF > 95. If the patient’s respiration was adequately maintained with BIS > 80 and TOF > 95, but they were unable to open their eyes on command (drowsy state, Ramsay sedation score above 3), flumazenil 0.3 mg was administered intravenously. The patient was observed in the Post-Anesthesia Care Unit for 40 min and transferred to a ward when the Modified Aldrete Score was >9. Pain was controlled using patient-controlled analgesia (PCA; fentanyl 500 μg, nefopam hydrochloride 100 mg, and ramosetron hydrochloride 0.3 mg) until postoperative day (POD) 3. All patients were discharged at POD 5 without complications.

### 2.4. Anesthetic Methods

For induction, 2 mg/mL remimazolam were infused at 6 mg/kg/h for 3 min regardless of apnea or loss of consciousness, and then, remimazolam was continuously infused at 1 mg/kg/h until the end of the surgery. Remifentanil was continuously administered, while the infusion rate was modified from 2 to 4 ng/mL by target-controlled infusion (TCI), which is a technique of intravenous anesthetic drug administration by a computer-controlled infusion pump based on the patient’s parameters, such as height, weight, age, and so on, depending upon blood pressure from induction to the end of surgery. When the patient achieved apnea, mask ventilation was performed concurrently, and endotracheal intubation was performed with the administration of rocuronium 0.8 mg/kg after 3 min of induction. The depth of anesthesia was monitored using BIS. During induction, the BIS target was set below 60 to ensure sufficient unconsciousness for general anesthesia. Throughout the surgery, the BIS score was maintained below 60 to ensure adequate anesthesia, while avoiding overly deep anesthesia by keeping the BIS score above 40. Anesthesia was maintained by these two agents together with O_2_ 1.5 L and air 2.5 L as fresh gas. At the end of the surgery, the infusion of both drugs was stopped simultaneously. No other anesthetic agents were used, except for an additional intravenous injection of 20 mg rocuronium when the tourniquet was deflated. When the operation was completed, sugammadex, 2~4 mg/kg, was injected intravenously.

### 2.5. Clinical Variables

#### 2.5.1. Primary Outcome

The requirement for rescue flumazenil administration in the operating room was the primary outcome of this study. Flumazenil was administered to patients who were treated with remimazolam when it was necessary to reverse excessive sedation or when emergence from anesthesia was delayed. The indications for administering flumazenil include failure to adequately regain consciousness, excessive sedation that poses a risk of respiratory depression, or prolonged sedation not resolving within a reasonable interval after completion of the remimazolam infusion [15]. The decision to administer flumazenil was made by attending anesthesiologists who were not involved in the study, ensuring an unbiased approach to patient management.

#### 2.5.2. Secondary Outcomes

The secondary outcomes of this study focused on the induction and recovery phases of general anesthesia. During induction, the occurrence of apnea and the time to apnea onset were assessed, along with the achievement of BIS < 60, indicating an adequate depth of anesthesia. The time required to reach this BIS was also recorded. In terms of recovery, the achievement of BIS > 80, which signifies a return to consciousness, was evaluated. The time required to reach this BIS was measured to compare recovery times between the younger and older patient groups.

### 2.6. Statistical Analyses and Sample Size

In our preliminary study, 24% of the <75 years group and 51% of the ≥75 years group needed flumazenil. Using an effect size of 0.5, a power (1–β) of 80%, and a significance level (α) of 0.05, the appropriate sample size for each group was calculated to be 50 participants. Considering a dropout rate of 10%, the sample size was adjusted to 55 participants per group. Therefore, 110 participants (55 per group) were required for this study.

The data are presented as the mean ± standard deviation for continuous variables and number (percentage) for categorical variables. The two age groups (<75 and ≥75 years) were compared using independent *t*-tests for normally distributed continuous variables. Categorical variables were analyzed using the chi-square test or Fisher’s exact test, depending on the expected cell counts. A *p*-value < 0.05 was considered statistically significant. All statistical analyses were performed using SPSS ver. 22 (IBM, Armonk, NY, USA); the figures were generated using Microsoft Excel 2021 (Microsoft, Redmond, WA, USA).

## 3. Results

### 3.1. Demographic Characteristics

#### 3.1.1. Preoperative Characteristics

The study included 99 patients divided into two groups based on age: the old group (*n* = 52) and the elderly group (*n* = 47) (Table 1). A significant difference in mean age was observed between the groups, with the old group averaging 68.8 ± 3.3 years and the elderly group averaging 78.1 ± 3.0 years (*p* < 0.001). Sex distributions were similar, with slightly higher proportions of women in both age groups, with 96.2% in the old group and 93.6% in the elderly group (*p* = 0.666). The ASA physical classification also did not differ between the two groups (*p* = 0.244). The majority of patients in both groups were classified as ASA class II, with 94.2% in the old group and 100% in the elderly group. Only the younger group had patients classified as ASA class I (5.8%) (Table 1).

#### 3.1.2. Intraoperative Variables

No significant differences in intraoperative variables were observed between the two age groups. The mean operation duration was slightly longer in the old group (79.9 ± 22.4 vs. 74.4 ± 12.9 min), but this difference was not statistically significant (*p* = 0.131). Similarly, mean anesthesia durations were comparable between the two groups, at 112.0 ± 23.8 vs. 107.2 ± 16.2 min (*p* = 0.25) in the old and the elderly groups. The tourniquet time and total remifentanil dosage were also similar in both age groups, with mean tourniquet times of 47.2 ± 14.6 vs. 48.6 ± 15.4 min (*p* = 0.647) and remifentanil dosages of 0.5 ± 0.2 vs. 0.4 ± 0.2 mg (*p* = 0.177) in the old and the elderly groups, respectively.

#### 3.1.3. Postoperative Variables

Postoperatively, the total PCA infusion was similar in both age groups, with mean total PCA infusions of 48.0 ± 21.6 vs. 47.8 ± 25.6 mL (*p* = 0.973) in the old and the elderly groups, respectively.

### 3.2. Remimazolam-Related Perioperative Variables

A comparison of remimazolam-related perioperative variables between the two age groups revealed no significant differences (Table 2). The intraoperative remimazolam infusion rate was slightly higher in the old group (23.0 ± 5.3 vs. 21.5 ± 5.5 μg/kg/min; *p* = 0.2). The requirement for flumazenil infusion in the operating room was slightly higher in the elderly group (38.3% vs. 25.0%; *p* = 0.154). No additional flumazenil was administered during the recovery period.

Apnea occurred in 76.9% of the old group and 85.1% of the elderly group, but the difference was not statistically significant (*p* = 0.302; Figure 2). The times to apnea onset also were comparable, with mean onset times of 99.8 ± 14.0 vs. 100.2 ± 19.9 s in the old and the elderly groups, respectively (*p* = 0.923). BIS < 60 was achieved in 82.7% of the old and 74.5% of the elderly groups (*p* = 0.318; Figure 2); the times to achieve this BIS reduction were 316.3 ± 486.5 vs. 358.8 ± 469.8 s, respectively (*p* = 0.698).

Both groups had similar outcomes in terms of regaining BIS > 80, with 82.7% vs. 83.0% in the old vs. the elderly groups, respectively (*p* = 0.970; Figure 3). The times to BIS recovery also were similar in the two groups, with 277.7 ± 234.2 vs. 267.7 ± 214.9 s, respectively (*p* = 0.842).

## 4. Discussion

There was no difference in the response to remimazolam, neither for anesthetic induction (apnea, achievement BIS < 60) nor for emergence (achievement BIS > 80, flumazenil use) between the old and the elderly patients, and the use of flumazenil for awakening was not increased in the elderly patients. This suggested that increasing age has little effect on the difference in the response to remimazolam.

Benzodiazepines, short-acting GABA_A_ receptor agonists, are widely used for sedation and general anesthesia due to their minimal cardiovascular effects. They provide hemodynamic stability, making them suitable for patients with hypovolemia or unstable hemodynamics. The ultra-short duration of action of the benzodiazepine remimazolam makes it particularly beneficial for short procedures and for patients with higher risks of hemodynamic or respiratory complications [16,17,18,19]. One report suggested that the optimal remimazolam dose for elderly patients is much lower than the usual dose for elderly patients [10]. However, we previously found no differences in anesthetic induction and emergence between age groups among patients administered a standard dose of remimazolam [20]. This result suggests that remimazolam can be used effectively in elderly patients without requiring dose adjustments based on age-related pharmacokinetic and pharmacodynamic differences.

Remimazolam has a superior safety profile to other anesthetics in elderly patients, with a lower incidence of hypotension and less pharmacodynamic variability due to its short action [21]. Consistent with this profile, no cases of hypotension, bradycardia requiring intervention, or other hemodynamic disturbances were observed in our study. Although remimazolam generally causes less respiratory depression [22], caution is warranted when it is combined with opioids such as remifentanil. This combination can lead to airway-related adverse events [23]. In our study, there was no significant difference in the occurrence of apnea between the two age groups, with apnea occurring in >75% of both groups. However, this occurrence did not contribute to any adverse respiratory events, supporting the use of remimazolam as a well-tolerated anesthetic, comparable to propofol, barbiturates, ketamine, or inhalational agents.

Apnea occurred within approximately 120 s in our study, which is consistent with domestic phase 3 clinical trial findings (HNP-2001; October 2019). The onset time did not exceed 2 min in either group, aligning with previous studies that considered a loss of consciousness for longer than 2 min to be clinically significant [24]. These results suggest that age does not accelerate the onset of apnea, and remimazolam remains appropriate for general anesthesia in elderly patients.

Our patients required 5–6 min to achieve a BIS < 60, and there were no instances of a BIS < 40 throughout the study. This is consistent with reports that maintaining an anesthetic depth > 40 promotes rapid recovery while reducing the risks of postoperative delirium and cognitive dysfunction [25,26,27].

Furthermore, the time to reach BIS > 80, indicating awakening, was ≤ 5 min, which was shorter than the 7 min threshold considered clinically significant in previous reports [28]. This finding demonstrates that remimazolam does not delay recovery with increasing age. Thus, it is comparable to other anesthetic agents.

Flumazenil, a benzodiazepine antagonist and negative allosteric modulator, was required at similar rates in both groups, suggesting that the need for flumazenil does not increase according to age at the same infusion rate and remifentanil dosage, implying minimal age-related impacts on delayed arousal.

There are various opinions regarding postoperative cognitive dysfunction (POCD), including that it shows positive effects [29,30] and that the effects cannot be confirmed [31,32]. In our study, there was no POCD development in both groups during postoperative hospital days. This appeared to be less associated with the occurrence of early POCD in old and elderly patients.

As global life expectancy increases, with the elderly population projected to constitute over 8.2% of the global population by 2025 according to the World Health Organization (WHO), and life expectancy in South Korea reaching 82.7 years in 2022 according to the Korean Statistical Information Service (KOSIS), it is important to reassess the use of anesthetics in elderly patients. Our findings challenge the conventional wisdom that anesthetic doses must be reduced with age, demonstrating that remimazolam is not significantly affected by age and, thus, allowing more flexible use in various clinical settings. Remimazolam is well-tolerated during induction, awakening, and recovery from anesthesia; it can be used safely in elderly patients, regardless of age.

A recent report showed that age plays an independent role in determining the appropriate dose of remimazolam for a proper hemodynamic response, with a recommendation to reduce the dose for older individuals compared to younger ones [33]. While age did not appear to be a significant factor influencing the difference in response to remimazolam in patients over 65 years in this study and limited to 65 to 85 years, it could not be confirmed with other age groups. In the future, it is necessary to conduct studies to investigate broadly the age-specific response to remimazolam by classifying the age groups into 18 to 64 years, 65 to 74 years, and >75 years.

This study had some limitations. It did not include patients aged > 85 years due to Institutional Review Board guidelines, which require legal guardian consent for this vulnerable population. As a result, our findings may not represent the effects of remimazolam in patients aged > 85 years. Furthermore, we did not evaluate hemodynamic responses as clinical variables because no critical adverse events (e.g., hypotension, bradycardia, or arrhythmias) requiring intervention occurred in our study cohort. This exclusion may limit the generalizability of our findings regarding the safety profile of remimazolam in broader clinical settings.

## 5. Conclusions

The requirement for flumazenil was similar in both groups, indicating that remimazolam is well-tolerated in elderly patients undergoing orthopedic surgery, with minimal age-related differences. Intra- and postoperative variables, such as infusion rate and anesthesia induction characteristics, including the time to apnea and changes in BIS, were consistent across age groups. Recovery characteristics, including the time to BIS recovery and the achievement of full consciousness, also were comparable among all geriatric patients. These findings suggest that remimazolam is a suitable anesthetic for elderly patients. However, future studies should explore its effects (including hemodynamic changes) across a broader age range of elderly populations.

## Figures and Tables

**Figure 1 medicina-60-01616-f001:**
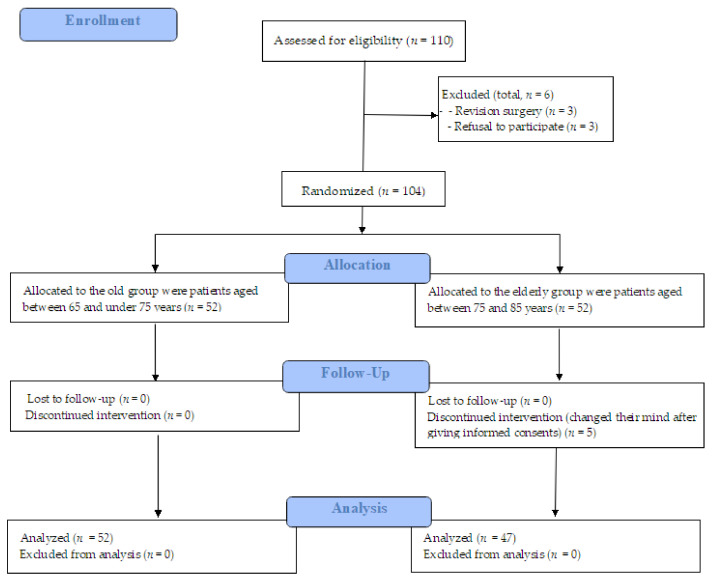
Flow diagram.

**Figure 2 medicina-60-01616-f002:**
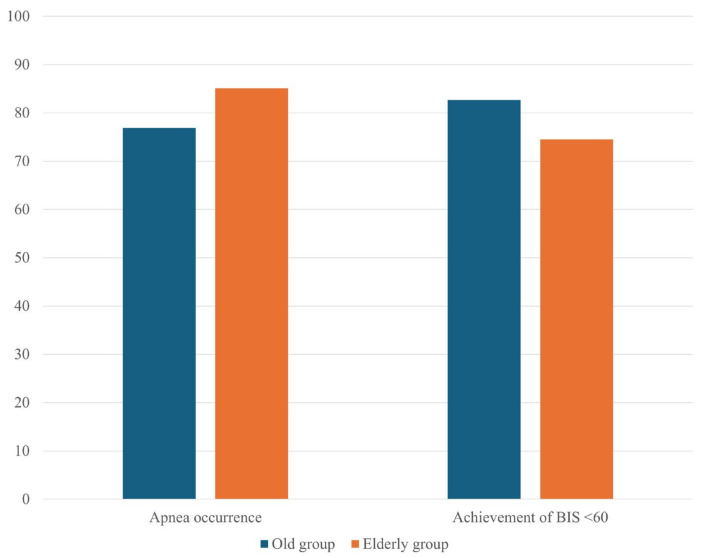
Comparison of apnea occurrence and achievement of BIS < 60 during general anesthesia induction between the ‘old group’ (65 to <75 years) and the ‘elderly group’ (75 to 85 years). Values are presented as percentages.

**Figure 3 medicina-60-01616-f003:**
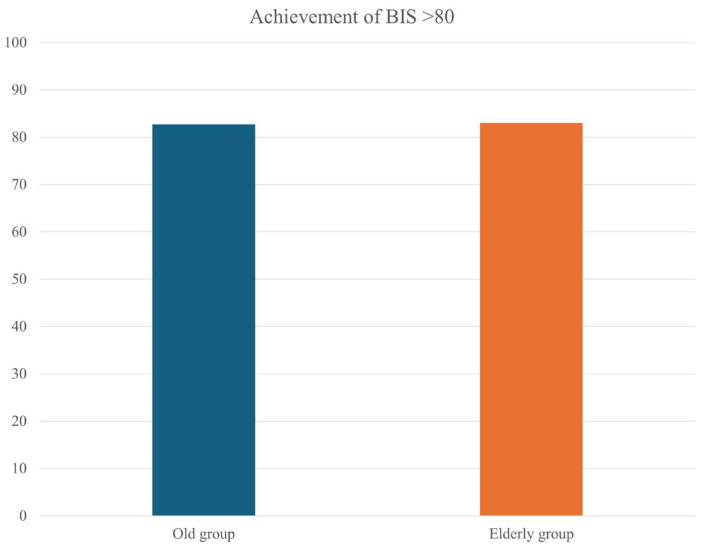
Comparison of achievement of BIS > 80 upon awakening from general anesthesia between the ‘old group’ (65 to <75 years) and the ‘elderly group’ (75 to 85 years). Values are presented as percentages.

**Table 1 medicina-60-01616-t001:** Comparison of demographic characteristics between between the old group (65 to <75 years) vs. the elderly group (75 to 85 years).

Group	Old Group	Elderly Group	*p*-Value
*n*	52	47	
** *Preoperative variables* **			
Age (years)	68.8 ± 3.3	78.1 ± 3.0	<0.001
Sex			0.666
Male	2 (3.8%)	3 (6.4%)	
Female	50 (96.2%)	44 (93.6%)	
ASA physical class			0.244
I	3 (5.8%)	0 (0.0%)	
II	49 (94.2%)	47 (100.0%)	
Body mass index (kg/m^2^)	25.2 ± 3.8	24.5 ± 2.5	0.288
Height (m)	1.6 ± 0.1	1.5 ± 0.1	0.332
Weight (kg)	61.4 ± 9.2	58.9 ± 7.0	0.145
** *Intraoperative variables* **			
Operation duration (min)	79.9 ± 22.4	74.4 ± 12.9	0.131
Anesthesia duration (min)	112.0 ± 23.8	107.2 ± 16.2	0.25
Tourniquet time (min)	47.2 ± 14.6	48.6 ± 15.4	0.647
Remifentanil			
Total dosage (mg)	0.5 ± 0.2	0.4 ± 0.2	0.177
** *Postoperative variables* **			
Total PCA infusion (mL)	48.0 ± 21.6	47.8 ± 25.6	0.973

Abbreviations: ASA, American Society of Anesthesiologists; PCA, patient-controlled analgesia. Values are expressed as mean (standard deviation) and number (percentage).

**Table 2 medicina-60-01616-t002:** Comparison of remimazolam-related perioperative variables between the old vs. the elderly groups.

Group	Old Group	Elderly Group	*p*-Value
*n*	52	47	
** *Intraoperative remimazolam* **			
Infusion rate (ug/kg/min)	23.0 ± 5.3	21.5 ± 5.5	0.2
Total infusion time (min)	101.0 ± 22.7	93.4 ± 17.5	0.054
** *Primary outcome* **			
flumazenil required in the OR	13 (25.0%)	18 (38.3%)	0.154
** *Secondary outcomes* **			
** *1. Induction of general anesthesia* **			
Apnea occurrence	40 (76.9%)	40 (85.1%)	0.302
Time to apnea onset (s)	99.8 ± 14.0	100.2 ± 19.9	0.923
Achievement of BIS < 60	43 (82.7%)	35 (74.5%)	0.318
Time to BIS < 60 (s)	316.3 ± 486.5	358.8 ± 469.8	0.698
** *2. Awake variables in general anesthesia* **			
Achievement of BIS > 80	43 (82.7%)	39 (83.0%)	0.97
Time to BIS > 80 (s)	277.7 ± 234.2	267.7 ± 214.9	0.842

Abbreviations: BIS, bispectral index; OR, operating room; Values are expressed as mean (standard deviation) and number (percentage).

## Data Availability

The data supporting the study findings are considered within the article.

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
