# Peer review of "Age-Related Response to Remimazolam among Older Patients Undergoing Orthopedic Surgery: A Single-Center Prospective Observational Study"

_medicina, 2024, doi:10.3390/medicina60101616_

Round 1

Reviewer 1 Report

Comments and Suggestions for Authors

The authors present a good review and observational study on the "Age-Related Response to Remimazolam among Older Patients Undergoing Orthopedic Surgery."

The methodology and  sample size and power appear adequate.

In the methods section,  I suggest that the authors indicate that with this target controlled infusion (TCI) technique as  some readers may not be familiar with the term TCI and TCI is not used in the USA. The authors should also indicate that their technique is also termed as a total intravenous anesthesia (TIVA) technique. 

The authors have two groups, < 75 years and > 75 years. I suggest that the authors use another term instead of stating that the group < 75 years is a young group.

 In many age related studies, a young group is defined as < 65 years. Old as 65 to 74 years, and elderly as > 75 years. As the average age in the "young" group is 68.8 years, this would be an "old" group and not young.

In the discussion, the authors should discuss more the age related differences of patients < 65 years compared to their "old" and "elderly" groups.   

The authors state that for neuromuscular block reversal they used sugammadex 200 mg. Was 200 mg given for all patients? Why was the recommended sugammadex dose of 2 mg/kg not used?

Table 1, the authors report BMI. Please also list patient's weight and height.

Table 1. The authors list the patient's ASA physical class as only 1 and 2. Old and elderly patients often have numerous co- morbidities. In South Korea, are the elderly patients so healthy that there are no ASA physical class 3 patients? Please explain.

Table 2.

What were the average and lowest BIS readings?  How did you define adequate anesthesia and amnesia? Is this a BIS between 60 and 80?

In your discussion , please discuss the effect of remimazolam on postoperative cognitive dysfunction (POCD) in elderly patients.

I think that the authors have done a fairly good job, but that their manuscript can be improved by further clarifications as noted.

Because the age difference (10 years) between the two groups (68 vs 78 years) are fairly close, I would not expect the results to be that different as the authors have shown. A better study would have been if they had three age groups: 18 to 64 years; 65-74 years, and > 75 years.

Author Response

Thank you for your valuable suggestion.
We have answered what you mentioned as follows.
The revised parts are highlighted in red.

Comment 1: In the methods section, I suggest that the authors indicate that with this target-controlled infusion (TCI) technique as some readers may not be familiar with the term TCI and TCI is not used in the USA. The authors should also indicate that their technique is also termed as a total intravenous anesthesia (TIVA) technique. 

Response 1: Remimazolam was not proceeded by TCI. Only remifentanil was proceeded with TCI application, This part has been revised in the manuscript with explanation of TCI technique.  (line 125-126)

Comment 2: The authors have two groups, < 75 years and > 75 years. I suggest that the authors use another term instead of stating that the group < 75 years is a young group.

 In many age-related studies, a young group is defined as < 65 years. Old as 65 to 74 years, and elderly as > 75 years. As the average age in the "young" group is 68.8 years, this would be an "old" group and not young.

Response 2: Thank you for your insightful comment regarding the age classification in our study.
We recognize that many studies define age groups differently, with < 65 years typically considered "young," 65–74 years as "old," and ≥ 75 years as "elderly." In our study, we originally categorized patients into two groups: < 75 years (termed "young" geriatric) and ≥ 75 years ("old" geriatric), focusing on comparing responses among older populations undergoing total knee arthroplasty. To better align with conventional age classifications used in similar studies, we have renamed the groups as the 'old group' (65 to < 75 years) and the 'elderly group' (75 to 85 years) based on what you mentioned.

We also agree that using the term "young" may not have accurately reflected common age classifications, especially since the average age in the < 75 years group is 68.8 years, which is more consistent with what is typically defined as an "old" group in other studies. To address this, we will update the terminology in the manuscript to accurately reflect this group as "old.". Additionally, in the Discussion section, we will further explore how the responses of patients aged < 65 years may differ from those in the "old" (65–74 years) and "elderly" (75-85 years) groups (line 283-290). While our study focused on patients aged 65 and older, we will also acknowledge relevant literature on younger populations (< 65 years) and the potential age-related differences in anesthesia response.

We appreciate your valuable suggestion and will incorporate these changes to ensure clarity and alignment with standard age classifications in the broader literature.

Comment 3: In the discussion, the authors should discuss more the age-related differences of patients < 65 years compared to their "old" and "elderly" groups.   

Response 3: we added ‘age related differences of patients’ in the discussion part as you mentioned. (line 283-290)

Comment 4: The authors state that for neuromuscular block reversal they used sugammadex 200 mg. Was 200 mg given for all patients? Why was the recommended sugammadex dose of 2 mg/kg not used?

Response 4:  As you mentioned, sugammadex is usually used at 2~4mg/kg. And if it is not continuous infusions, 2mg/kg is recommended. In this study, considering the weight of the patients, the predicted sugammadex dose is 100~200mg. However, 20mg of rocuronium was added at the time of torniquet deflation, and the time until the end of the surgery did not exceed 30 minutes after additional rocuronium administration, it makes necessary to more than 2mg/kg. In addition, 200 mg of sugammadex administration is our institutional protocol, so we routinely apply it regardless of weight in adult patients.

Based on this, as you mentioned, we have addressed this concern by presenting a modified dose in the revised manuscript to clearly convey the meaning (Line 137).

Comment 5: Table 1, the authors report BMI. Please also list patient's weight and height.

Response 5: I confirmed. As requested, we have included the patients' weight and height in Table 1 to provide a more complete representation of their demographic characteristics (page 5).

Comment 6: Table 1. The authors list the patient's ASA physical class as only 1 and 2. Old and elderly patients often have numerous co- morbidities. In South Korea, are the elderly patients so healthy that there are no ASA physical class 3 patients? Please explain.

Response 6: Thank you for your insightful comment. In this study, the inclusion criteria were intentionally designed to select patients with relatively low anesthetic risk (ASA physical class I and II). This was done to minimize the variability introduced by severe comorbidities, which could confound the outcomes related to the effects of remimazolam on anesthesia and recovery in elderly patients.

While ASA physical class III (uncontrolled DM or unstable angina and so on) patients are also common among the elderly population in South Korea, we excluded them from this study to focus on a more homogenous group of relatively healthy elderly patients. This approach allows us to better isolate the effects of remimazolam without the potential influence of severe underlying health conditions. Future studies could certainly address this by including a wider range of ASA classes to evaluate remimazolam effects on patients with higher anesthetic risks. We addressed this concern page 3 (line 82~89).

Comment 7: Table 2. What were the average and lowest BIS readings?  How did you define adequate anesthesia and amnesia? Is this a BIS between 60 and 80?

Response 7: Thank you for your question regarding the BIS readings and our definition of adequate anesthesia and amnesia. In both the 'old group' and 'elderly group,' the average BIS readings during surgery were consistently maintained within a safe range, with the lowest recorded values remaining above 40. This range is considered appropriate for ensuring adequate anesthesia without risking excessive depth or delayed recovery.

In our study, we defined adequate anesthesia as achieving and maintaining a BIS score below 60 during the induction phase, a standard threshold indicating that the patient is sufficiently unconscious for surgery. Conversely, a BIS score of 80 was used during the recovery phase to signify the patient had regained consciousness. Thus, the BIS score range of 60 to 80 played a key role in different phases of anesthesia management: 60 for ensuring adequate depth during induction and 80 for confirming recovery. Maintaining BIS above 40 was crucial to avoid overly deep anesthesia, which could delay emergence and increase the risk of postoperative cognitive issues.

We hope this explanation clarifies your concerns. Please let us know if you require any further information. We addressed this concern as follows (page 4; line 130-133)

Comment 8: In your discussion, please discuss the effect of remimazolam on postoperative cognitive dysfunction (POCD) in elderly patients.

Response 8: None of the patients developed POCD in this study.

We have addressed this in the discussion section as you mentioned. (page 8; line 278-282)

Comment 9: I think that the authors have done a fairly good job, but that their manuscript can be improved by further clarifications as noted.

Because the age difference (10 years) between the two groups (68 vs 78 years) are fairly close, I would not expect the results to be that different as the authors have shown. A better study would have been if they had three age groups: 18 to 64 years; 65-74 years, and > 75 years.

Response 9: Thank you for your good opinion. As we planned to study elderly patients, the age distribution of the study subjects has a limitation. In our institution, TKRA is only performed on those over 60 years of age, and the research subjects were selected based on the criteria of 65 years of age or older, which is the age defined as geriatrics. In future studies, we will proceed by dividing the age groups as you mentioned. We addressed this in discussion section, page 9, line 297-299

Reviewer 2 Report

Comments and Suggestions for Authors

Dear authors,

Thank you for submitting your work.

I find a discrepancy in the title and the type of study. The title mentions a prospective observational study, while this is an RCT. Please revise this.

Also, address the comments as in the document attached, make the changes, and resubmit.

Thanks.

Comments on the Quality of English Language

The English usage is satisfactory in the concerned submission.

Author Response

Comment 1: I find a discrepancy in the title and the type of study. The title mentions a prospective observational study, while this is an RCT. Please revise this.

Response 1: Thank you for your observation.

Since the study groups were divided based on age, this is not a randomized controlled trial (RCT), but rather a prospective observational study.
I modified the text (line 73).

Comment 2: Also, address the comments as in the document attached, make the changes, and resubmit.

Response 2: We sincerely appreciate your careful review and comments.
We have revised what you mentioned in the manuscript and provided the answers.
The revised parts are highlighted in blue.

The line here represent the line from the revised version manuscript.

  • Line 73: I confirmed. I modified this on the revised version. This study was a prospective observational study.
  • Line 97: I confirmed. To convey the meaning clearly, we changed ‘the exclusion’ to ‘the inclusion’ as you mentioned.
  • Line 107: I confirmed. We added this to the separate subheading 2.4. anesthetic methods as you mentioned (line 120~138).
  • Line 232: I confirmed. We added the key results findings to the beginning of the Discussion part as you mentioned. (line 232~236)
